# Subcritical Water Enhanced with Deep Eutectic Solvent for Extracting Polysaccharides from *Lentinus edodes* and Their Antioxidant Activities

**DOI:** 10.3390/molecules27113612

**Published:** 2022-06-04

**Authors:** Jixian Zhang, Zhiqiang Ye, Guoyan Liu, Li Liang, Chaoting Wen, Xiaofang Liu, Youdong Li, Tao Ji, Dongming Liu, Jiaoyan Ren, Xin Xu

**Affiliations:** 1College of Food Science and Engineering, Yangzhou University, Yangzhou 225127, China; zjx@yzu.edu.cn (J.Z.); yezhiqiang1104@163.com (Z.Y.); liugy@yzu.edu.cn (G.L.); liangli0508@hotmail.com (L.L.); chaoting@yzu.edu.cn (C.W.); liuxf@yzu.edu.cn (X.L.); liyoudonghh@163.com (Y.L.); jt2216297499@163.com (T.J.); donming533223@163.com (D.L.); 2School of Food Science and Engineering, South China University of Technology, Guangzhou 510641, China

**Keywords:** subcritical water, deep eutectic solvent, response surface, structure, antioxidant activities

## Abstract

In the present study, subcritical water extraction (SWE) assisted with deep eutectic solvent (DES) is used to extract *Lentinus edodes* polysaccharides (LEP). In addition, the antioxidant activity of the polysaccharide samples was also investigated. Based on a single factor test and response surface test, the optimal extraction factors were a liquid–solid solvent of 40:1 mL/g, extraction temperature of 147.23 °C, water content of 39.76% and extraction time of 17.58 min. Under these extraction conditions, the yield of LEP was 6.26 ± 0.08%. Compared with the SWE and hot water extraction (HWE), it improved by 19.24% and 17.01%, respectively. In addition, the results of monosaccharide composition, molecular weight, FT-IR, UV and SEM confirmed that the extracts had the features of polysaccharides. Interestingly, the polysaccharides obtained with the SWE assisted with the DES procedure showed a higher DPPH scavenging activity, hydroxyl radical scavenging activity and hydrogen peroxide scavenging activity, which indicated that the polysaccharides with this method had a stronger antioxidant activity. These findings demonstrated that the SWE-assisted DES is a strong method to obtain polysaccharides from *Lentinus edodes* for food, biopharmaceutical and other industrial production.

## 1. Introduction

*Lentinus edodes* is a nutrient-rich edible fungus. *Lentinus edodes* is mainly distributed in an arc on the west side of the Pacific Ocean and is famous all over the world for its unique nutritional value. *Lentinus edodes* are rich in protein, polysaccharides, purines, choline and other biologically active ingredients [1,2,3]. Due to the presence of these nutrients, *Lentinus edodes* has various physiological activities, such as antitumor, antiviral, antibacterial and hypotensive activities [4,5]. Among these ingredients, polysaccharides were noticed in the field of medicine and healthcare due to their immunostimulatory activity, antioxidant activity, and antitumor activity, among others [6,7]. Traditionally, the methods widely used for polysaccharides extraction are hot water extraction (HWE), alkali solution extraction and polyethylene glycol extraction, etc. [8,9]. However, these methods have the disadvantages of higher energy consumption, time consuming, lower efficiency and environmental pollution [10]. Based on above analysis, the extraction method of polysaccharides is becoming diversified [11]. Some novel extraction technologies have emerged one after another, such as microwave-assisted extraction (MAE), ultrasonic-assisted extraction (UAE), subcritical water extraction (SWE), and ultrahigh pressure extraction (UPE), among others [12,13,14]. Notably, polysaccharides extracted by different methods showed diversity in structure and activity [15]. Especially, SWE method has attracted more attention due to their great advantages, such as environmental protection, energy saving and high efficiency, among others [16].

Subcritical water is when the ambient temperature of the water between 100 °C and 374 °C and the ambient pressure is 1–22.1 MPa [17]. In the subcritical state, the dielectric constant of water decreases with increasing temperature. As the temperature increases from 100 °C to 374 °C, the dielectric constant decreases from 80 to 27, and the polarity of water is similar to that of alcohol at room temperature [18]. Therefore, the active ingredients with different polarities can be continuously extracted. Additionally, the thermal and mechanical effects of subcritical water can improve the extraction efficiency by destroying the cell wall in the matrix materials and improving the mass and heat transfer effect [19]. At present, SWE was a novel extraction procedure to obtain bioactive substances, such as polysaccharides [20,21]. For example, compared with the extraction rate of UE (3.6 ± 0.31%) and HWE (5.2 ± 0.50%) methods, the SWE method exhibited a higher extraction rate (13.15 ± 1.05%) of polysaccharides, because of the enhanced cell destruction capacity, capillary effect and solvent penetration [22]. Besides, Gu et al. found that compared with the extraction rate of HWE (12.99 ± 0.23%), SWE can increase the polysaccharide extraction rate of *Sagittaria sagittifolia* L. (20.37 ± 0.54%) and significantly improve their antioxidant activity and immunomodulatory activities [23]. Interestingly, to further promote the dissolution of polysaccharides, many studies added a deep eutectic solvent to further improve the extraction rate of polysaccharides. DES is a new type of organic solvent of the binary or ternary system composed of hydrogen bond donor and hydrogen bond acceptor [24], which has the advantages of easy preparation, biocompatibility and environmental friendliness [25]. These solvents have been widely used to extract biomacromolecules, such as polysaccharides, lignin and pectin [26]. Some studies have confirmed that SWE assisted with DES could help to further improve the extraction efficiency of polysaccharides. For example, Collin et al. extracted Solomon Islands red seaweed polysaccharide by SWE-assisted ionic liquid, and the result showed that extraction rate of polysaccharides significantly improved (*p* < 0.05) compared with pretreatment with the CaCl_2_ and HWE method [15]. However, there is no report on SWE enhanced with DES for extracting polysaccharides from *Lentinus edodes*. 

Therefore, the process of the SWE-assisted DES extraction of LEP is optimized by single factor and response surface experiments. Then, the structure of the obtained polysaccharides is initially identified and compared with the polysaccharides extracted with HWE and SWE. Finally, the differences in the activity of polysaccharides obtained with different methods are investigated by chemical antioxidant tests in vitro.

## 2. Materials and Methods

### 2.1. Materials and Reagents

The *Lentinus edodes* were obtained from the Chahe Vegetable Market in Yangzhou City, China. Choline chloride, Malonate, Urea, Glycerol, 1,4-Butanediol, 1,3-Butanediol, Ethylene glycol and 1,2-Propanediol were purchased from Aladdin reagent Co., Ltd., (Shanghai, China). Glucose, galactose, xylose, mannose and rhamnose were purchased from Shanghai Yuanye Biological Technology Co., Ltd., (Shanghai, China). 1,1-diphenyl-2-picrylhydrazyl (DPPH) was obtained from Sigma-Aldrich (St. Louis, MO, USA). Other reagents were of analytical grade and obtained from Sinopharm Reagent Co., Ltd., (Shanghai, China).

### 2.2. The Pretreatment of Lentinus Edodes

The *Lentinus edodes* dried at 60 °C for 24 h with a constant temperature blast dryer. After drying, the sample was pulverized by a pulverizer and passed through a 50-mesh screen to remove large particles. The moisture content of *Lentinus edodes* is 0.56%. The raw material powder was degreased by Soxhlet extraction with petroleum ether as a reagent for 6 h to obtain defatted powder for the subsequent experiment.

### 2.3. Preparation of DESs

The synthesis method of DESs was described by Chen et al. [27] and modified according to the actual synthesis situation. The step to make DESs was prepared by mixing hydrogen bond donors and choline chloride in specific molar ratios (1:2). The mixture of two reagents and deionized water were stirred in 85 °C environment to obtain clear and transparent solution. The solvents were stored at ambient temperature. The types of DESs compared in this experiment are shown in Table 1.

### 2.4. SWE Assisted with DES for Extracting Polysaccharides

The autoclave (KH, Shanghai Yuhua Instrument Co., Ltd., Shanghai, China) was used to react in the subcritical state, and the volume used in this experiment was 50 mL. Briefly, *Lentinus edodes* powders were pretreated under the presupposed extraction time (min), temperature (°C), liquid–solid solvent (mL/g) and water content (%). After that, the extracts were added 4 times the volume of ethanol and stored at 4 °C for 12 h. The precipitate was obtained by centrifugation and reconstituted with deionized water and deproteinized with Sevage reagent [28]. The solution was dialyzed through 3500 cut-off dialysis bag for 2 days to remove small molecules. The last step was freeze-drying to obtain the final sample.

In this experiment, the phenol-sulfuric acid method was used to measure the total sugar content of crude polysaccharide samples [29]. A formula *Y* (%) = *c*/*w* × 100% was used to calculate the yield of polysaccharides, where *c* represents the weight of polysaccharides and *w* represents the weight of the dried extracts.

### 2.5. Single Factor Design of Experiments

Single-factor experiments determine the medium value of the factor’s optimal range. The temperature (150 °C), extraction time (15 min), liquid–solid solvent (30 mL/g) and water content (50%) were set as the experimental condition of single-factor. When one-factor changes, the other three remain fixed values.

### 2.6. BBD Design of Experiments 

After the completion of the single-factor experiment, the response surface method (RSM) was used to optimize the range of the factors. A Box–Behnken design (BBD) was performed based on the 4 independent factors at 3 levels in this experiment. This experiment examined four factors (*X*_1_: extraction temperature, *X*_2_: extraction time, *X*_3_: liquid–solid ratio, and *X*_4_: water content) to achieve the best extraction conditions of polysaccharides. The response value of the single-factor experiments with polysaccharide extraction rate (*Y*) are shown in Table 2. To reduce the mutual influence between variables, the experiment was performed at random.

The variables were coded according to Equation (1):(1)$$X_{i}=\frac{A_{i}-A_{0}}{\Delta A}$$
where *A_i_* is the (dimensionless) coded value of the variable *X_i_*, *A*_0_ is the value of *A_i_* at the center point, and Δ*A* is the step change.

Four independent factors of the Box–Behnken design (BBD) (*X*_1_: extraction temperature, *X*_2_: time, *X*_3_: liquid–solid ratio, and *X*_4_: water content) were the main variables in the experiment. The consequence of the optimization was shown in Table 3. In the optimization process, the responses are related to factors by quadratic models, which also include the linear model as shown in Equation (2).
(2)$$Y=A_{0}+\sum_{i=1}^{4}A_{i}X_{i}^{2}+\sum_{i=1}^{4}A_{ii}X_{i}^{2}+\sum_{i=1}^{3}\sum_{j=i+1}^{4}A_{ij}X_{i}X_{j}$$
where *Y* represent the extraction rate of polysaccharide, *A*_0_ is a constant coefficient, and *A_i_*, *A_ii_*, and *A_ij_* are coefficients estimated by the model. *X_i_* and *X_j_* are the levels of the independent variables. These parameters represent the linear, quadratic and cross-product effects of the *X*_1_, *X*_2_, *X*_3_ and *X*_4_ factors on the response, respectively. The model evaluates the influence of different independent variables on the response value. All results in this experiment were analyzed by the Design Expert (version 8.0.1) (Stat-Ease, Minneapolis, MN, USA).

### 2.7. Comparison between Different Extraction Methods

#### 2.7.1. Hot Water Extraction

*Lentinus edodes* powders and deionized water were mixed in a specific liquid–solid ratio (30 mL/g) and stirred at 80 °C for two hours. The further purification process of the polysaccharide solution was the same as in Section 2.3. The polysaccharide obtained in this way was named LEPH.

#### 2.7.2. The Extraction of Subcritical Water 

In the experiment of extracting polysaccharides from *Lentinus edodes* powder by subcritical water method, the temperature was set at 140 °C, and the liquid–material ratio was set at 25 mL/g; the extraction time was 20 min and the extraction pressure is 1.5 MPa. The further purification process of the polysaccharide solution was the same as in Section 2.3. The polysaccharide obtained in this way was named LEPS.

#### 2.7.3. The Extraction of SWE Assisted with DES

The sample extracted by SWE assisted with the DES method were obtained under optimal polysaccharide extraction conditions. The polysaccharide obtained in this way was named LEPD.

### 2.8. Structural Characteristics

#### 2.8.1. Monosaccharide Composition 

The determination of monosaccharide composition was performed according to the previous method with some modifications [30]. The samples were complete hydrolyzed with 2 M H_2_SO_4_ (5 mL). After hydrolysis, the reaction solution was neutralized by BaCO_3_ powders. Then, the supernatant was rotary evaporated to obtain a dry powder. The next step was to add hydroxylamine hydrochloride of the same quality and 0.5 mL pyridine to the dry sample. The heating reaction in 95 °C water baths was for 30 min. The alditol acetate derivative was formed after adding the anhydride (0.5 mL). The GC analysis was analyzed on the Agilent 7890A Gas Chromatograph (Palo Alto, Santa Clara, CA, USA) with an HP-5 column (30 m × 0.32 mm × 0.25 μm) and flame ionization detection (FID). The GC procedure was as follows: the initial column temperature was 130 °C for 5 min, the detector temperature was 300 °C, and then the temperature was raised to 240 °C and maintained in this state for 5 min. The injection volume is 2 μL. Standard samples were treated in the same way.

#### 2.8.2. Molecular Weight

The molecular weight of polysaccharides was determined by gel permeation chromatography (GPC) [31]. The injection concentration of the sample is 1 mg/mL. The molecular weight was measured on Shimadzu LC-20A prominence liquid chromatography (Kyoto, Japan), which was equipped with Shimadzu RID-20A detector and TSK gel G2000SWXL column. 

#### 2.8.3. Ultraviolet Spectroscopy

The samples were dissolved with deionized water to carry out UV spectroscopic detection, and the solution concentration was 0.1 mg/mL The ultraviolet spectroscopy was analyzed from 200 to 800 nm by Lambda 35 UV-Vis spectra photometer (PerkinElmer Ltd., Waltham, MA, USA).

#### 2.8.4. FT-IR Spectroscopy

The FT-IR spectroscopy was performed with an FT-IR spectrometer (Cary 670 FTIR, Agilent Technologies, Santa Clara, CA, USA) in the range of 4000–400 cm^−1^.

#### 2.8.5. SEM Analysis

Spread the polysaccharide sample evenly on the conductive glue pasted on the copper sample stage, then made it conductive by gold spraying operation to facilitate photographic observation. The Scanning Electron Microscope (SEM, Gemini SEM 300, Carl Zeiss, UK) was used to observe the microscopic morphology of the samples.

### 2.9. Antioxidant Activities

#### 2.9.1. DPPH Radical Scavenging Activity

The method to measure the DPPH free radical scavenging activity of samples has been reported previously [32]. The polysaccharides were diluted to different concentrations (1.0, 2.0, 3.0, 4.0 and 5.0 mg/mL). Then, the polysaccharides solution (2 mL) of different concentrations was mixed with 2 mL of the 0.2 mM DPPH ethanol solution. After that, the mixed solution was placed at room temperature and protected from light for 30 min and was tested at 517 nm on a UV1000 Spectrophotometer (Shanghai Tianmei Instrument Co., Ltd., Shanghai, China). The following equation was calculated for the DPPH radical scavenging rate (3):(3)$$\mathrm{DPPH}\;\mathrm{radical}\;\mathrm{scavenging}\;\mathrm{rate}\;(\%)=(1-\frac{A_{1}-A_{2}}{A_{0}})\;\times \;100$$
where *A*_0_ represents the absorbance of the control (the deionized water in replace of the sample), *A*_1_ represents the absorbance of the sample, and *A*_2_ represents the absorbance of the sample and ethanol without DPPH.

#### 2.9.2. Hydroxyl Radical Scavenging Activity

The hydroxyl radical scavenging activity of the polysaccharides was measured according to the reported method [33]. Briefly, 1 mL of 9 mM FeSO_4_ solution, 0.5 mL of 8.8 mM H_2_O_2_, 1 mL 9 mM salicylic acid and 1 mL of ample at different concentration (1.0, 2.0, 3.0, 4.0, 5.0 mg/mL) were mixed and incubated at 37 °C for 30 min. The mixture was incubated together at 37 °C for 30 min. The absorbance of the samples was measured at 510 nm. The hydroxyl radical scavenging activity was expressed as Equation (4):(4)$$\mathrm{Hydroxyl}\;\mathrm{radical}\;\mathrm{scavenging}\;\mathrm{rate}\;(\%)=(1-\frac{A_{1}-A_{2}}{A_{0}})\;\times \;100$$
where *A*_0_ is the absorbance of the solution without sample, *A*_1_ is the absorbance of the sample, and *A*_2_ is the absorbance of the reagent blank without salicylic acid. 

#### 2.9.3. H_2_O_2_ Scavenging Activity

The H_2_O_2_ scavenging activity of the sample was measured according to the previous method with minor modifications [34]. Briefly, 1.0 mL 10 mmol/L H_2_O_2_ (prepared with pH7.4 phosphate buffer) and 1.0 mL of the polysaccharides at different concentrations (1.0, 2.0, 3.0, 4.0, 5.0 mg/mL) were mixed well. The absorbance of the mixture was recorded at 248 nm. The H_2_O_2_ scavenging activity was expressed as the following Equation (5):(5)$$H_{2}O_{2}\;\mathrm{scavenging}\;\mathrm{rate}\;(\%)=(1-\frac{A_{1}-A_{2}}{A_{0}})\;\times \;100$$
where *A*_0_ represents the absorbance of the solution without sample, *A*_1_ represents the absorbance of the sample, and *A*_2_ represents the absorbance of the blank.

### 2.10. Statistical Analysis

All results were performed by using SPSS version 26.0 (SPSS Inc., Chicago, IL, USA) data processing software to determine that the experimental results are statistically significant. The analysis of BBD used the Design-Expert software trial version 8.0.1 (Stat-Ease, Minneapolis, MN, USA) to perform regression analysis and graph optimization.

## 3. Results and Discussion 

### 3.1. Different Types of DES Selection

The properties of the extraction solvent (diffusion, solubility, surface tension, viscosity, polarity, etc.) directly affect the permeability of the extraction solvent to the raw material, which in turn affects the efficiency of extraction [35]. From Figure 1, in terms of extraction rate, DES-2 has a better performance than other DES reagents. The possible reason is that DES-2 has a higher hydrogen bond and the ability to interact with more hydrogen bonds than other DES reagents [36]. Therefore, DES-2 was selected to extract the LEP.

### 3.2. Single-Factor Experiment

#### 3.2.1. The Factor of Extraction Temperature

The progress of the SWE process is highly affected by temperature [37]. High temperature can quickly destroy the structure of raw materials and extract biological macromolecules by reducing the viscosity of the extraction solvent and improving the fluidity of the solvent [38]. As can be seen from Figure 2A, the yield of polysaccharides gradually increased as the extraction temperature increased from 110 °C to 150 °C, while the temperature continued to increase the yield of polysaccharides gradually showed a downward trend. This phenomenon can be explained by the fact that high temperature increases the solubility of polysaccharides in the extraction solvent and promotes the diffusion of target substances out of the cells [39]. However, when the ambient temperature is higher than a certain limit, the structure of the polysaccharide may be damaged or even degraded. Therefore, the extraction temperature was selected from 130 °C to 170 °C.

#### 3.2.2. The Factor of Extraction Time

The picture in Figure 2B shows that the yield of polysaccharides increased with the extraction time (5–15 min) and decreased from 15 min. This was might be due to the fact that with the increase in extraction time, the polysaccharide is degraded due to the influence of the high temperature [40]. Therefore, the factor of time was selected at 10–20 min for the next experiment.

#### 3.2.3. The Factor of the Liquid–Solid Ratio

The liquid–solid ratio is also an important factor in the process of extraction. It can be seen from Figure 2C that the liquid–solid ratio increased from 10:1 to 30:1 mL/g, and the yield of polysaccharides increased simultaneously. However, when the liquid–solid ratio exceeds 30:1 mL/g, the response began to fall sharply. This phenomenon shows that when the liquid–solid ratio increases, the polysaccharides are more easily leached from materials [41]. The yield of polysaccharides significantly reduced, when the liquid–solid ratio was beyond 30:1 mL/g. This was due to the fact that the solute was saturated in the solvent and had a negative impact on the mass transfer efficiency, thereby reducing its yield [42]. At the same time, excessive use of solvents can also cause solvent waste. Therefore, the liquid–solid ratio of 20:1 to 40:1 mL/g was selected to carry out the next experiment.

#### 3.2.4. The Factor of Water Content

The water content affects the leaching ability of the solvent [43]. From Figure 2D, in the process of the water content increase from 10% to 50%, the polysaccharide yield increased rapidly, while the yield decreased when the water content exceeds 50%. This could be because the increase in water content can increase the dissolution rate of polysaccharides, while the viscosity of DES decreases with the higher water content. The decrease in viscosity leads to a decrease in the leaching ability and a decrease in the extraction effect of polysaccharides [44]. The water content of DES ranging from 30% to 70% was selected.

### 3.3. The Extraction of LEPD Optimized by BBD

#### 3.3.1. Statistical Analysis and Model Fitting

Model building results are judged by variance (ANOVA) analysis. Table 4 showed that the *p*-value was significant (*p* < 0.0001) and the *F*-value was 28.08, which indicates that the mathematical model of this experiment is significant. The lack of fit is not significant (*p* > 0.1411), indicating that the regression equation can explain the results well and predict the optimal conditions. In addition, the linear coefficients (*X*_1_, *X*_2_, *X*_3_, and *X*_4_), cross-product coefficients (*X*_1_*X*_2_, *X*_1_*X*_4_, *X*_2_*X*_4_, and *X*_3_*X*_4_) and quadratic coefficients (*X*_1_^2^, *X*_2_^2^, and *X*_4_^2^) exhibited significant effect (*p* < 0.05), whereas the other coefficients (*X*_1_*X*_3_, *X*_2_*X*_3_, and *X*_3_^2^) were not significant (*p* > 0.05). Additionally, the determination coefficient (*R*^2^ = 0.9656, $R_{adj}^{2}$ = 0.9312, and C.V.% = 4.41) confirmed the model has a high accuracy and can predict 96.56% of the experimental results. Through the *F*-value test, the order of the influence of the four factors on the extraction process of LEPD is as follows: *X*_2_ (Time) > *X*_3_ (Liquid–solid solvent) > *X*_4_ (Water content) > *X*_1_ (Temperature).

The 3D response surfaces and 2D contour plots are used to show the relationship between independent and dependent variables; the interaction between the two variables and the effect on LEP yield are shown in Figure 3. Figure 3A,E demonstrates the effect of the interaction between temperature and time on polysaccharide yield. When the liquid–solid ratio and water content were constant, the extraction rate of polysaccharides increased with the increase in extraction time and temperature. Figure 3B,F shows how time and water content affect the polysaccharide yield with temperature and the liquid–solid ratio at a specific value. Interestingly, the yield of polysaccharides was positively correlated with water content and temperature first and then negatively correlated. It can be seen from the above situation that, in order to obtain an optimal extraction rate, the optimum conditions for temperature and moisture content must be set. Figure 3C,G presented the effect of time and liquid–solid ratio on the yield of polysaccharides. From the two-dimensional contour, the promotion effect of the liquid–solid ratio on extraction rate was stronger than that of time. The interaction of the two factors was manifested in the increase in the extraction rate. Figure 3D,H presents the effect of water content and time on the yield of polysaccharides. When the liquid–solid ratio and the temperature were constant, the water content can promote the dissolution of polysaccharides within a small range while the temperature was rising, which manifested that the water content had a greater influence on the dissolution of polysaccharides. The figures showed the basis to select the optimal reaction conditions.

As shown in Table 3, the yield ranged from 3.23% to 5.94%. According to the multiple regression analysis of the test data, the following equations were constructed from the results for the independent variables and response values: *Y* (%) = 0.25*X*_1_ + 0.61*X*_2_ + 0.29*X*_3_ − 0.26*X*_4_ − 0.27*X*_1_*X*_2_ − 0.21*X*_1_*X*_3_ + 0.32*X*_1_*X*_4_ + 0.23*X*_2_*X*_3_ − 0.31*X*_2_*X*_4_ − 0.15*X*_3_*X*_4_ − 0.47*X*_1_^2^ − 1.04*X*_2_^2^ + 0.025*X*_3_^2^ − 0.64*X*_4_^2^
where *Y* represents the yield of the polysaccharides, and *X*_1_, *X*_2_, *X*_3_ and *X*_4_ represent the temperature (°C), time (min), liquid–solid ratio (g/mL) and water content (%), respectively. 

#### 3.3.2. Verification of the Model

The optimum extraction conditions for polysaccharides optimized by Design-Expert software are the liquid–solid ratio of 40:1 mL/g, extraction time of 17.58 min, the water content of DES 39.76% and the extraction temperature of 147.23 °C. Under the optimal extraction conditions, the yield of LEPD is 6.26 ± 0.08%, which was similar to the 6.36% predicted by the software. The results show that the model was established successfully. Furthermore, the extraction rate of LEPD improved by 19.24% and 17.01% compared with LEPS (5.25 ± 0.05%) and LEPH (5.35 ± 0.11%), respectively. The model is suitable for optimizing the extraction method of LEP.

### 3.4. Comparison of LEPD with Other Samples

#### 3.4.1. Monosaccharide Composition Analysis

The GC analysis results of different polysaccharides were shown in Figure 4. The consequence of the standard products that the LEPD consisted of D-galactose (7.0%), D-glucose (90.3%) and D-mannose (2.7%), and the molar ratio of 1:32.90:2.54 (Figure 4B). The LEPS consisted of D-galactose (17.4%), D-glucose (73.3%), and D-mannose (9.3%), and with a molar ratio of 1:7.85:1.86 (Figure 4C). Besides, the composition of LEPH is D-galactose (13.5%), D-glucose (78.5%), and D-mannose (8.0%), with a molar ratio of 1:9.77:1.68 (Figure 4D). These results can conclude that the basic structures of the three polysaccharides are similar in monosaccharide composition, but have different molecular molar ratios. These polysaccharides are all heteropolysaccharides. A previous study found that the monosaccharide of polysaccharides obtained from *Lentinus edodes* by HWE was only glucose (100%) [45]. From the monosaccharide composition results, the method of SWE assisted with DES can change the basic structure of the extracted polysaccharide and may affect the activity of the polysaccharide. From the above study, it is found that the monosaccharide composition of polysaccharides is affected by raw materials and extraction methods [46].

#### 3.4.2. Analysis of Molecular Weight

The bioactivity of polysaccharides is significantly influenced by molecular weight [47]. Some studies showed that the LEP with medium molecular weight has better biological activity [48]. The molecular weights of LEPD, LEPS and LEPH are 6.3 × 10^4^ Da, 6.6 × 10^4^ Da and 6.4 × 10^4^ Da, respectively (Figure 5A). There was no significant effect on the molecular weight of polysaccharides among these methods.

#### 3.4.3. UV Analysis

The absorption spectra (200–400 nm) of the three polysaccharides are shown in Figure 5B. UV spectroscopy results show that the sample has no absorption at 280 nm and 260 nm, which indicated the absence of proteins and nucleic acids in these samples [49].

#### 3.4.4. FT-IR Analysis

FT-IR spectroscopy is an extremely important method for polysaccharide structure analysis, which can characterize the chemical structure of polysaccharides at a specific wavenumber [50]. The FT-IR spectra of polysaccharides are depicted in Figure 5C. Signals at 3300, 2920, 1620, 1400, and 1100 cm^−1^ were the typical characteristic of polysaccharides [51,52]. The strong absorption peak near 3300 cm^−1^ comes from the stretching of the O-H group. The signal of 2922 cm^−1^ is caused by the C-H stretching vibration. The strong absorptions at around 1600 cm^−1^ indicated the stretching of C=O. The absorption near 1400 cm^−1^ was due to the deformation vibration of C-H [53]. The signals of the C-O-C groups generally appear around 1000–1100 cm^−1^, indicating that these three polysaccharides are pyranose rings [54]. The weaker signal near 890 cm^−1^ represents the vibration of the β-glycosidic bond, while there is no absorption peak around 847 cm^−1^, indicating that there is no α-glycosidic bond in the polysaccharides [55]. The results of the IR spectrum indicated that LEPD, LEPS, and LEPH have a similar structure and are composed of pyranose linked by β-glycosides. This is consistent with the fact that neither the deep eutectic solvent nor the subcritical water have an effect on the base groups of the sample [20,56].

#### 3.4.5. SEM Analysis

To compare the difference in the appearance of polysaccharide samples, the polysaccharides were scanned using SEM. The microtopography of SEM is shown in Figure 6. The LEPD has a looser structure, rough and porous surface, and a high degree of fragmentation (Figure 6A). The LEPS has a leaf-like surface shape, and uneven distribution but with a smooth and fine surface, indicating a strong intermolecular interaction between molecules (Figure 6B). The LEPH sample shows an overlapping adhesion sheet structure, and the sample aggregates into irregular compact clusters (Figure 6C). Due to being affected by the extraction method, the surface morphology and structure of the obtained polysaccharides show large differences [57]. The different extraction methods of polysaccharides may lead to some differences in the apparent morphology of polysaccharides [58]. 

### 3.5. Antioxidant Activities of LEPs 

#### 3.5.1. DPPH Scavenging Activity

DPPH is a stable radical with maximum absorbance at 517 nm [59]. When DPPH free radicals encounter proton-donating substances (such as antioxidants), the free radicals are scavenged and absorbance decreases [60]. The consequence of DPPH scavenging activity was shown in Figure 7A. The ability of the three polysaccharides to scavenge DPPH free radicals showed a concentration-dose effect (1.0–5.0 mg/mL). The IC_50_ values of LEPD, LEPS and LEPH sample scavenging activity were 0.551 mg/mL, 0.9 mg/mL and 1.37 mg/mL, respectively. The LEPD sample showed a better DPPH scavenging activity than LEPS and LEPH under the experimental concentrations. The difference in activity tests is related to structural changes. It suggested that changes in polysaccharide structure caused by SWE assisted with DES can improve the DPPH radical scavenging capacity of polysaccharides.

#### 3.5.2. Hydroxyl Radical Scavenging Activity

Hydroxyl free radicals cause cell death in the human body, because it is a kind of reactive oxygen species [61]. When the hydroxyl radical is scavenged by proton-donating substance groups, such as antioxidants, their absorbance at 510 nm decreases [62]. The consequences of hydroxyl scavenging activity are presented in Figure 7B. The IC_50_ value of polysaccharides for hydroxyl radical scavenging were 2.013 mg/mL, 3.009 mg/mL and 2.871 mg/mL, respectively. From the results, we can conclude that the LEPD sample has a stronger hydroxyl scavenging activity. The experimental results are similar to the experimental results using an only deep eutectic solvent or subcritical water to extract polysaccharides [63,64].

#### 3.5.3. Hydrogen Peroxide Scavenging Activity

Hydrogen peroxide has a high cell membrane permeability and cytotoxicity; this substance has a strong toxic effect on the cells of the organism [65]. The consequence of hydrogen peroxide is presented in Figure 7C. The scavenging activity of LEPD, LEPS, and LEPH increased with their concentration increases (1–5 mg/mL). From the experimental results, we can conclude that LEPD has a better hydrogen peroxide scavenging ability compared with LEPS and LEPH. In conclusion, the LEPD have better antioxidant activity among the three polysaccharides. The molecular weight and monosaccharide composition of the different samples may have contributed to this result [66].

## 4. Conclusions

The SWE assisted with DES is a new method to obtain polysaccharides from *Lentinus edodes*. The extraction conditions were optimized by BBD as follows: temperature of 147.23 °C, the liquid–solid ratio of 40:1 mL/g, the water content of 39.76%, and time of 17.58 min. Under the optimal conditions, the extraction rate of polysaccharides is 6.26 ± 0.08%, which improved by 19.24% and 17.01% compared with SWE (5.25 ± 0.05%) and HWE (5.35 ± 0.11%), respectively. In addition, the polysaccharides with the different extraction methods have obvious differences in monosaccharide composition, molecular weight, and apparent morphology. Moreover, LEPD has a higher capacity in scavenging DPPH free radical, hydroxyl radical, and hydrogen peroxide, which exhibited higher antioxidant activities. Therefore, the extraction method of SWE assisted with DES can be used to extract substances from other materials. At the same time, this method can change the structure of the polysaccharide and thereby affect the activity of the polysaccharide. It is foreseeable that this will be a better method for extracting highly active polysaccharides.

## Figures and Tables

**Figure 1 molecules-27-03612-f001:**
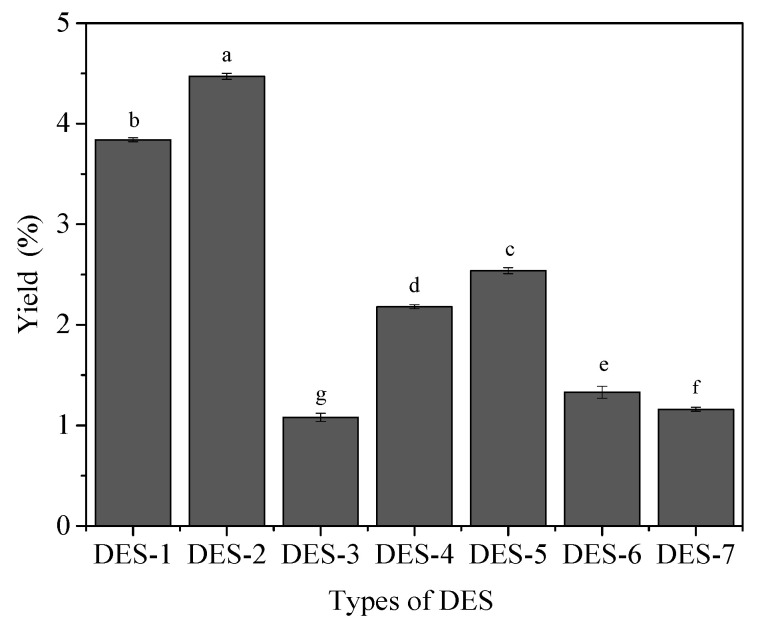
The extraction rate of different DES solvents. a–g indicated that there were significant differences (*p* < 0.05).

**Figure 2 molecules-27-03612-f002:**
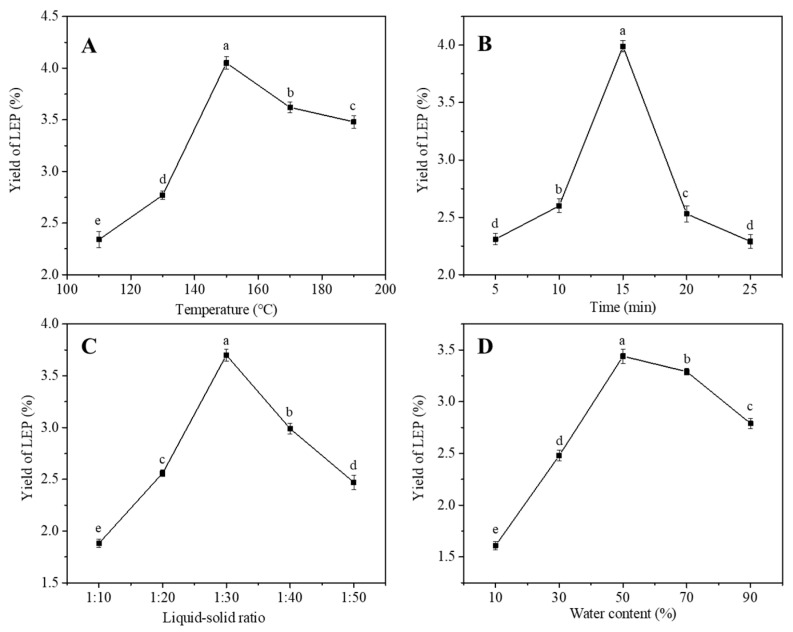
Effect of temperature (**A**), time (**B**), liquid–solid ratio (**C**) and water content (**D**) on the extraction yield of polysaccharides. Superscripts (a–e) indicated that the results are significantly different (*p* < 0.05).

**Figure 3 molecules-27-03612-f003:**
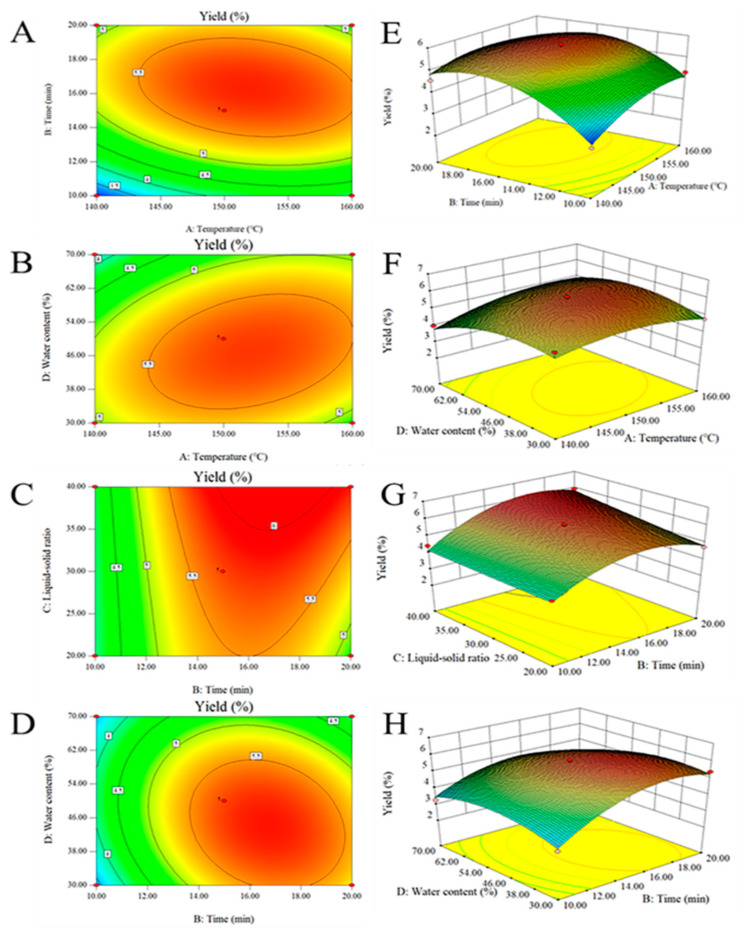
Contour plots (**A**–**D**) and response surface plots (**E**–**H**).

**Figure 4 molecules-27-03612-f004:**
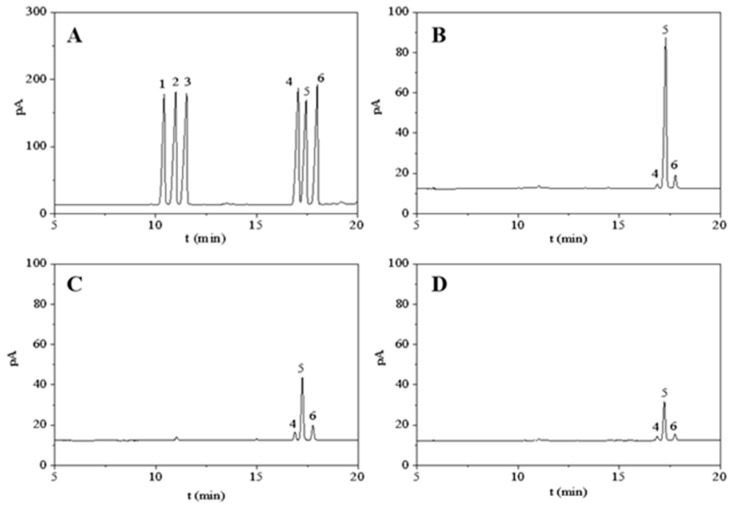
Results of the monosaccharide composition of monosaccharide standard (**A**), LEPD (**B**), LEPS (**C**), and LEPH (**D**). Peaks 1–6 represented L-rhamnose, D-arabinose, D-xylose, D-mannose D-glucose and D-galactose, respectively.

**Figure 5 molecules-27-03612-f005:**
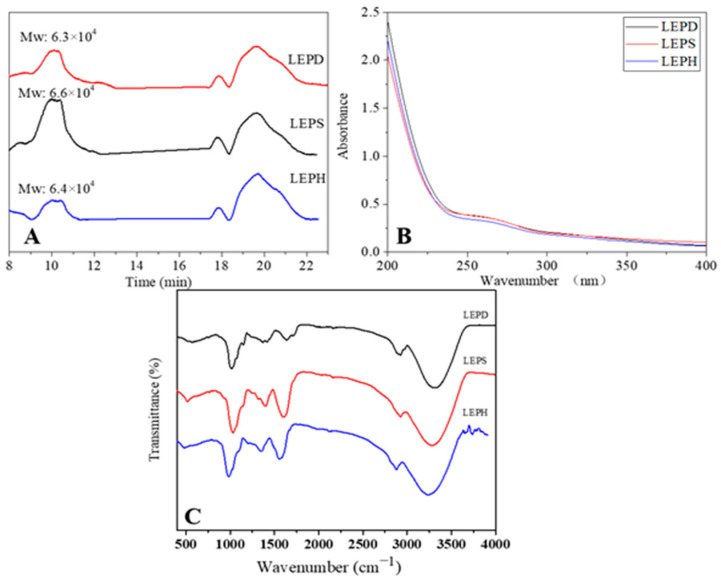
The molecular weight result of different polysaccharides (**A**), UV spectrum of the polysaccharides (**B**), and IR spectrum of the polysaccharides (**C**).

**Figure 6 molecules-27-03612-f006:**
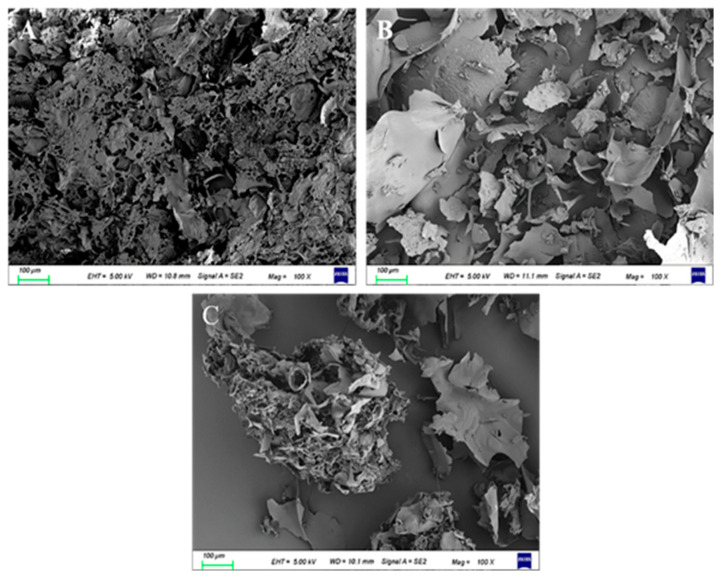
Results of LEP extracted by different extraction methods with 100× SEM. LEPD (**A**), LEPS (**B**), and LEPH (**C**).

**Figure 7 molecules-27-03612-f007:**
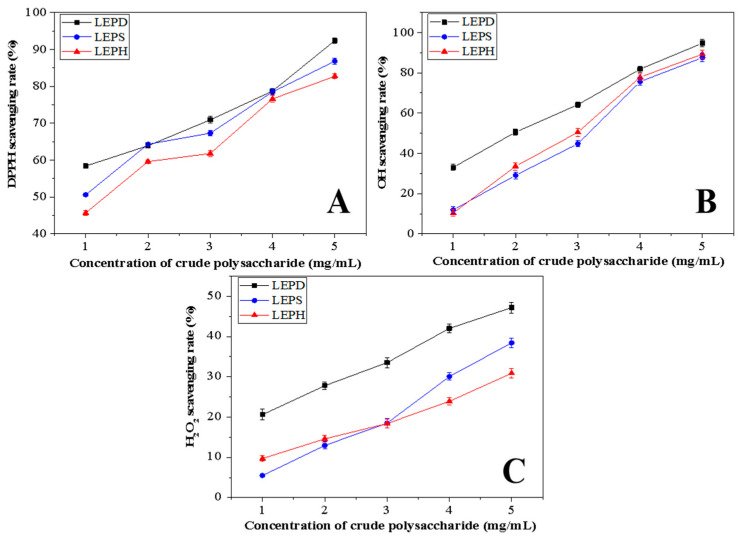
Antioxidant activity of LEPD, LEPS, and LEPH in vitro. The scavenging rates of (**A**) DPPH, (**B**) OH, and (**C**) H_2_O_2_ of LEPD, LEPS and LEPH at different concentrations.

**Table 1 molecules-27-03612-t001:** Different types of DESs used in this experiment.

Number	Component 1	Component 2	Molar Ratio
DES-1	Choline chloride	Urea	1:2
DES-2	Choline chloride	Malonate	1:2
DES-3	Choline chloride	1,2-Propanediol	1:2
DES-4	Choline chloride	Glycerol	1:2
DES-5	Choline chloride	Ethylene glycol	1:2
DES-6	Choline chloride	1,3-Butanediol	1:2
DES-7	Choline chloride	1,4-Butanediol	1:2

**Table 2 molecules-27-03612-t002:** The independent variables of BBD.

Variable	Levels		
	−1	0	+1
Temperature (°C)/*X*_1_	140	150	160
Extraction time (min)/*X*_2_	10	15	20
Liquid–solid solvent (mL/g)/*X*_3_	20	30	40
Water content (%)/*X*_4_	30	50	70

**Table 3 molecules-27-03612-t003:** The result of Box–Behnken design.

Std.	No.	*X* _1_	*X* _2_	*X* _3_	*X* _4_	*Y* (%)
5	1	150	15	20	30	4.80 ± 0.13
6	2	150	15	40	30	5.68 ± 0.06
26	3	150	15	30	50	5.88 ± 0.08
29	4	150	15	30	50	5.69 ± 0.12
23	5	150	10	30	70	3.27 ± 0.05
15	6	150	10	40	50	4.43 ± 0.16
13	7	150	10	20	50	4.13 ± 0.08
7	8	150	15	20	70	4.72 ± 0.06
11	9	140	15	30	70	3.99 ± 0.14
25	10	150	15	30	50	5.86 ± 0.11
17	11	140	15	20	50	4.60 ± 0.10
12	12	160	15	30	70	4.80 ± 0.08
1	13	140	10	30	50	2.93 ± 0.04
18	14	160	15	20	50	5.50 ± 0.16
3	15	140	20	30	50	4.56 ± 0.12
28	16	150	15	30	50	5.61 ± 0.17
2	17	160	10	30	50	4.30 ± 0.07
27	18	150	15	30	50	5.60 ± 0.09
20	19	160	15	40	50	5.48 ± 0.13
9	20	140	15	30	30	5.26 ± 0.07
19	21	140	15	40	50	5.40 ± 0.05
16	22	150	20	40	50	5.94 ± 0.12
21	23	150	10	30	30	3.23 ± 0.05
24	24	150	20	30	70	4.18 ± 0.09
10	25	160	15	30	30	4.77 ± 0.06
22	26	150	20	30	30	5.37 ± 0.11
8	27	150	15	40	70	5.01 ± 0.05
4	28	160	20	30	50	4.86 ± 0.16
14	29	150	20	20	50	4.74 ± 0.07

Note: *X*_1_, Temperature; *X*_2_, Extraction time; *X*_3_, Liquid–solid solvent; *X*_4_, Water content; *Y*: Yield of polysaccharides.

**Table 4 molecules-27-03612-t004:** ANOVA for response surface quadratic model.

Source	Sum of Squares	df	Mean Square	*F*-Value	*p*-Value Prob > *F*	
Model	17.93	14	1.28	28.08	<0.0001 ^a^	significant
X_1_	0.74	1	0.74	16.11	0.0013 ^a^	
X_2_	4.51	1	4.51	98.96	<0.0001 ^a^	
X_3_	0.99	1	0.99	21.74	0.0004 ^a^	
X_4_	0.82	1	0.82	18.01	0.0008 ^a^	
X_1_ X_2_	0.29	1	0.29	6.27	0.0252 ^a^	
X_1_ X_3_	0.17	1	0.17	3.69	0.0755 ^b^	
X_1_ X_4_	0.42	1	0.42	9.26	0.0088 ^a^	
X_2_ X_3_	0.20	1	0.20	4.44	0.0536 ^b^	
X_2_ X_4_	0.38	1	0.38	8.29	0.0121 ^a^	
X_3_ X_4_	0.087	1	0.087	1.91	0.1889 ^b^	
X_1_^2^	1.45	1	1.45	31.72	<0.0001 ^a^	
X_2_^2^	6.96	1	6.96	152.65	<0.0001 ^a^	
X_3_^2^	4.108 × 10^−3^	1	4.108 × 10^−3^	0.090	0.7685 ^b^	
X_4_^2^	2.69	1	2.69	58.90	<0.0001 ^a^	
Residual	0.64	14	0.046			
Lack of Fit	0.57	10	0.057	3.13	0.1411 ^b^	Not significant
Pure Error	0.072	4	0.018			
Cor Total *R*^2^ *R*^2^_adj_ *R*^2^_pred_ Adeq precision C.V.%	18.57 0.9656 0.9312 0.8182 17.914 4.41	28				

^a^ Significant (*p* < 0.05); ^b^ Not significant (*p* > 0.05).

## Data Availability

Data are contained within the article.

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
