# Peer review of "Subcritical Water Enhanced with Deep Eutectic Solvent for Extracting Polysaccharides from Lentinus edodes and Their Antioxidant Activities"

_molecules, 2022, doi:10.3390/molecules27113612_

Round 1

Reviewer 1 Report

I would recommend a few revisions:

Line 47. The reference is missing, or maybe it is Gereniu et al. (2018)

Line 109. The information is not necessary

Line 89. Please separate the sentences from table 2. It appears like a footnote

Line 140. Table 3. Are the results made in duplicate? The design does not ensure the repeatability by Runs 3,4,10, 16, and 18? What does std mean? No means the run?

Line 143. The model was applied for section 2.5, or the optimum extraction condition obtained by the model was applied here? If so, please specify that

Line 237. Please insert the need or explain the application of the single factor experiment (e.g., the selection parameters for the design model??). Provide some information in the material and methods section

Line 280. What LEPD mean?

 Lines 381-382. Move the sentence lower in the text

General remark. Please compare the results with other similar findings from the literature

Reviewer 2 Report

The comments are as follows.

  1. Please, revise the text to correct some misprints.
  2. The abbreviations should be defined just the first time they are mentioned in the text and used as such throughout the text.
  3. Please, provide more details regarding the plant material (manipulation after obtaining the material, moisture content, particle size).
  4. The extraction pressure is missing in the section 2.5.2. Please, provide it.
  5. The Results and discussion, and Conclusion sections might be improved by expressing more the importance of the SWE-DES application.
